# Synthesis and Biological Evaluation of Novel Furopyridone Derivatives as Potent Cytotoxic Agents against Esophageal Cancer

**DOI:** 10.3390/ijms25179634

**Published:** 2024-09-05

**Authors:** Xingyu Ren, Jiaojiao Zhang, Anying Dai, Pengzhi Sun, Yibo Zhang, Lu Jin, Le Pan

**Affiliations:** College of Chemistry and Chemical Engineering, Xinjiang Agricultural University, Urumqi 830052, China; 18299520272@163.com (X.R.); koh9391421@163.com (J.Z.); 13609979146@163.com (A.D.); sun8302587@163.com (P.S.); 18290824750@163.com (Y.Z.); lu_lu_jin@163.com (L.J.)

**Keywords:** furanopyridinone derivatives, synthesis and modification, antitumor activity

## Abstract

Cancer continues to be a major global health issue, ranking among the top causes of death worldwide. To develop novel antitumor agents, this study focused on the synthesis of a series of 21 novel furanopyridinone derivatives through structural modifications and functional enhancements. The in vitro anti-tumor activities of these compounds were investigated through the cytotoxicity against KYSE70 and KYSE150 and led to the identification of compound **4c** as the most potent compound. At a concentration of 20 µg/mL, compound **4c** demonstrated a remarkable 99% inhibition of KYSE70 and KYSE150 cell growth after 48 h. IC_50_ was 0.655 µg/mL after 24 h. Additionally, potential anti-tumor cellular mechanisms were explored through molecular docking, which was used to predict the binding mode of **4c** with METAP2 and EGFR, suggesting that the C=O part of the pyridone moiety likely played a crucial role in binding. This study provided valuable insights and guidance for the development of novel anticancer drugs with novel structural scaffolds.

## 1. Introduction

Cancer remains a pervasive and aggressive malady, ranking among the leading causes of mortality globally. It is distinguished by the unchecked proliferation of cells, which can spread from one organ to another via metastasis throughout the body [1,2]. Among the various types of cancer, esophageal cancer is considered to rank as the sixth most common mortality cancer type worldwide, which is prevalent worldwide [3,4]. Currently, chemotherapy remains as the main treatment method for various cancers with long-term proven clinical effects that can save many lives and extend the lives of many patients [5]. However, chemotherapy has the disadvantages of poor selectivity and non-targeted distribution [6]. Numerous plant-derived natural products have attracted considerable interest for their potential as antitumor drugs [7]. Consequently, the modification of natural products to uncover new anticancer agents has emerged as a potent strategy in the fight against cancer [8].

Furan rings have shown remarkable efficacy in selectively targeting tumor cells [9,10,11]. Some furan derivatives have exhibited a remarkable range of biological activities and played a crucial role in the molecular structure of pharmaceutical chemistry for anti-bacterial and anti-fungal applications, including furazolidone, furantanone, furantoin, and furacillin [12,13,14]. Meanwhile, the heterocyclic skeleton is an important structure for many natural products with excellent biological activity, and it is widely used in the design and discovery of new molecules with potential pharmacological activity. Pyridine-ring drugs are an extensive category of heterocyclic compounds, encompassing both natural drugs and synthetic drugs, like nicotine, piroxicam, quinine, camptothecin, diazepam, 4-vinylpyridine, aripiprazole, and lidocaine [15,16]. The addition of heterocyclic structures into drug molecules is an important way to design and synthesize biologically active compounds in anti-tumor drugs; as shown in Figure 1, some compounds of synthetic natural furopyridines and synthetic furopyridines have been published so far [17]. As a result, there has been a keen focus on the synthesis and isolation of furan-containing heterocyclic compounds, driven by their profound significance in both pharmaceutical and agricultural applications [18].

Furopyridine heterocyclic derivatives display differential drug activity in terms of their anticancer and antiviral effects [19]. A series of trifluoromethyl-substituted pyrido[3′,2′:4,5]furo[3,2-d]pyrimidine derivatives were synthesized by Bielenica in 2015 and two highly effective compounds were identified as anticancer agents against the neuro-2a cell line, with IC_50_ values of 5.8 and 3.6 µM, respectively [20]. In this study, a series of furan–pyridinone compounds were designed and synthesized from 3-furan-carboxylic acid, and their anticancer activities were investigated to obtain more promising compounds against KYSE70 and KYSE150 cell lines, with IC_50_ values of 0.655 µg/mL after 24 h. This study provided valuable insights and guidance for the development of novel anticancer drugs with novel structural scaffolds.

## 2. Results and Discussion

### 2.1. Chemistry

As shown in Figure 1, a series of furan–pyridine **2d**–**2f** derivatives were designed and synthesized by replacing the amino acid.

The compound 3-furoyl-L-leucine was synthesized by the Schotten–Baumann acylation reaction with L-leucine and 3-furoyl chloride. Then, the leading compound 6-isobutyl-5,6-dihydrofuro [3,2-c] pyridine-4,7-dione was synthesized by Friedel–Crafts acylation as an intramolecular cyclization reaction of 3-furoyl-L-leucine catalyzed by Eaton’s reagent. The furan [3,2-c] pyridine derivatives were reduced by NaBH_4_ from the parent compound. The furan [3,2-c] pyridine derivatives **3a**–**3k** were synthesized by reflux with acetonitrile based on parent compound **2b** and the potassium carbonate as a catalyst. The general reaction formula is shown in Figure 2.

The furan [3,2-c] pyridine derivatives **4a**–**4d** were synthesized by using compound **3e** as the nuclear compound, and the potassium carbonate as a catalyst under the acetonitrile reflux condition. The general reaction formula is shown in Figure 3.

### 2.2. Biological Activity

#### Cytotoxicity Activity

Novel furan pyridine derivatives were screened for their anti-tumor activities on two cancer cell lines (KYSE70 and KYSE150) using the MTT method.

The cytotoxic activities of novel furan pyridine derivatives against two cancer cells are shown in Table 1. The preliminary activity tests proved that compounds **3b**, **3e**, **3f**, **3i**, and **4c** had significant anti-tumor activity against KYSE70 cells at a concentration of 40 μg/mL. Meanwhile, compounds **3e** and **4c** also showed measurable anti-tumor activity against KYSE150 cells at a concentration of 40 μg/mL.

The results indicated that compound **3e** had moderate inhibitory activity against the two types of cancer cells mentioned above, which was chosen as the parent compound of compound **4a**–**c**. Also, compound **4c** was discovered with excellent anti-tumor activity, and exhibited an inhibitory activity of over 99% against KYSE70 and KYSE150 cells. On this basis, in order to further explore its anti-tumor activity, we investigated how the inhibitory activity of the cells was influenced by different concentrations of compound **4c** in treatment.

After conducting additional tests on compound **4c**, we established its IC_50_ value against KYSE70 cells to be 1.463 μg/mL after 24 h and 1.329 μg/mL after 48 h. As for KYSE150 cells, the IC_50_ value reached 0.888 μg/mL at 24 h and 0.655 μg/mL at 48 h. Additionally, we generated graphs depicting the inhibition rates of compound **4c** against KYSE70 and KYSE150 cells under various concentrations (detailed in Table 2).

In order to further study the anti-tumor activity of the compounds, compound **4c** was selected with an anti-tumor rate of more than 99%, its semi-maximum effect concentration (IC_50_) value was calculated, and is summarized in Table 3. The results showed that compound **4c** had excellent anti-tumor activity. Compound **4c** showed strong inhibitory effects against KYSE150, and its IC_50_ value was 0.655 μg/mL in 48 h.

For KYSE70 and KYSE150 cells, the compounds exhibited a similar trend in their anti-tumor activities, with compounds **3b** and **3e** exhibiting excellent activity. However, the anti-tumor activity of bromopropyl substituent (compound **3b**) against KYSE70 cells was stronger than that against KYSE150 cells. Similarly, isobutyl substituent (compound **3f**) exhibited stronger anti-proliferative activity against KYSE150 cells compared with KYSE70 cells. The above results indicate that, when compound **2b** was used as the precursor, the bromopropyl substituent could significantly enhance the inhibitory activity of the compound against KYSE70 cells, while isobutyl substituents could significantly enhance the inhibitory activity of the compound against KYSE70 cells.

The substitution of olefins could indeed significantly enhance anti-tumor activity, which is consistent with previous research results [21]. Moreover, as the number of olefin carbon atoms increases, the anti-tumor activity of the compounds increases. But when the number of carbon atoms in olefins increased to six (compound **3j**), the anti-tumor activity of the compounds decreased significantly. The substitution of maleimide substituent (compound **4c**) and succinimide substituent (compound **4d**) had a significant difference in the inhibition of the cell proliferation of the compounds. The maleimide substituent could significantly decrease the cell viability of the compounds, while succinimide substituents could have the opposite effect. This could be because the double bond on the pyrrole ring in the maleimide substituent forms a conjugated diene structure with two carbonyl groups, thereby changing the overall electron distribution of the compound and enhancing its anti-tumor activity. Related studies have found that maleimide can react with protein thiol groups, thereby affecting the protein function, which may be closely related to its activity.

### 2.3. Molecular Docking Assessment

To investigate the potential binding mechanisms of derivative **4c** with the EGFR (PDB ID: 6DUK) and MetAP2 (PDB ID: 5D6E) proteins, a molecular docking study was conducted. These human proteins were selected as key targets for the study of esophageal cancer due to their recognized roles in promoting the proliferation of esophageal cancer cells.

As shown in Figure 2 and Figure 3, the molecular docking of the highly active compound **4c** with two receptor proteins is depicted in 2D and 3D diagrams. Compound **4c** docked with the receptor protein 5D6E, as shown in Figure 2. The furan ring of the parent compound was involved in a π-stacking interaction with amino acid residue Trp148. The carbonyl group of the pyridinone on the parent compound formed hydrogen bonds with amino acid residues Arg149 and Arg144, respectively. Additionally, a carbonyl group in the substituent formed a hydrogen bond with amino acid residue Lys155. These results suggest that the protein METAP2 could be a potential target for furan–pyridinone derivatives, with the C=O part of the pyridone moiety likely playing a crucial role in binding.

Furthermore, in Figure 3, compound **4c** is shown docked with the receptor protein EGFR. The furan ring of the parent compound is involved in π-stacking with amino acid residue Trp880. The carbonyl group of the pyridinone on the parent compound forms hydrogen bonds with the amino acid residues Arg841 and Lys913, respectively. A carbonyl group in the substituent forms a hydrogen bond with amino acid residue Arg841, and another carbonyl group in the substituent forms a hydrogen bond with amino acid residue Ala722. These findings indicate that the protein EGFR could also be a potential target for furan–pyridinone derivatives, with the C=O part of the pyridone moiety being an important interaction point.

### 2.4. Spectral Data ***2a–f***, ***3a–k*** and ***4a–b***

More information can be found in Appendix A.

N-(3-furancarbonyl)-L-leucine (**2a**)

Yield: 27.5%; m.p. 66.2–67.1 °C; ^1^H NMR (600 MHz, methanol-*d*_4_) δ 7.90 (t, *J* = 1.2 Hz, 1H), 7.57 (t, *J* = 1.7 Hz, 1H), 6.68 (dd, *J* = 2.0, 0.9 Hz, 1H), 3.19 (s, 3H), 3.06 (s, 3H), 1.82 (dtd, *J* = 9.2, 7.1, 4.4 Hz, 1H), 1.41 (dqd, *J* = 14.2, 7.3, 3.8 Hz, 1H), 1.20 (ddd, *J* = 12.8, 9.0, 7.4 Hz, 1H), 0.99 (s, 1H), and 0.98–0.79 (m, 6H). ^13^C NMR (150 MHz, methanol-*d*_4_) δ 166.88, 145.40, 143.79, 121.96, 108.31, 50.70, 48.02, 47.88, 47.74, 47.60, 47.45, 47.31, 47.17, 39.95, 24.78, 22.01, and 20.31.

6-isobutyl-furano[3,2-c]pyridine-4,7-dione (**2b**)

Yield: 40.3%; m.p. 63.4–64.5 °C; ^1^H NMR (600 MHz, methanol-*d*_4_) δ 8.08 (s, 2H), 7.54 (s, 2H), 6.83 (s, 2H), 4.59 (dd, *J* = 10.2, 4.4 Hz, 2H), 4.49–4.44 (m, 2H), 4.42 (s, 1H), 4.19–4.12 (m, 1H), 3.68 (s, 1H), 3.28 (s, 11H), 1.77–1.63 (m, 16H), 1.25 (d, *J* = 17.5 Hz, 1H), 0.95 (dd, *J* = 17.0, 5.7 Hz, 3H), and 0.90 (s, 6H). ^13^C NMR (150 MHz, methanol-*d*_4_) δ 176.17, 165.33, 146.80, 145.20, 123.36, 109.72, 57.47, 57.33, 52.09, 41.36, 27.98, 26.19, 23.41, 21.73, and 17.28.

7-hydroxy-6-isobutyl-furan[3,2-c]pyridin-4(5H)-one (**2c**)

Yield: 63.5%; m.p. 58.4–60.1 °C; ^1^H NMR (600 MHz, Methanol-*d*_4_) δ 7.72–7.67 (m, 1H), 7.15 (t, *J* = 1.8 Hz, 1H), 6.43 (d, *J* = 2.0 Hz, 1H), 4.44 (s, 2H), 4.22 (dd, *J* = 10.3, 4.3 Hz, 1H), 3.31 (s, 1H), 2.90–2.89(m, 1H), 1.39–1.19 (m, 4H), and 0.55 (dd, *J* = 12.7, 5.7 Hz, 7H). ^13^C NMR (150 MHz, methanol-*d*_4_) δ 174.75, 165.35, 146.87, 145.24, 123.22, 109.69, 52.69, 52.21, 41.14, 26.12, 23.30, and 21.74.

N-(3-furancarbonyl)-L-isoleucine (**2d**)

Yield: 32.4%; m.p. 71.2–72.1 °C; ^1^H NMR (600 MHz, methanol-*d*_4_) δ 8.13–8.10 (m, 1H), 7.54 (t, *J* = 1.8 Hz, 1H), 6.84 (d, *J* = 1.8 Hz, 1H), 4.47 (d, *J* = 6.5 Hz, 1H), 3.30 (s, 2H), 1.95 (dtd, *J* = 9.1, 6.8, 4.4 Hz, 1H), 1.57 (dqd, *J* = 15.0, 7.5, 4.2 Hz, 1H), 1.26 (ddd, *J* = 13.5, 9.0, 7.2 Hz, 1H), 0.96 (s, 1H), and 0.99–0.82 (m, 6H). ^13^C NMR (150 MHz, methanol-*d*_4_) δ 175.00, 165.27, 146.84, 145.17, 123.29, 109.81, 70.52, 58.33, 49.57, 38.08, 26.55, 16.04, 14.46, and 11.62.

6-(sec-butyl)-furo[3,2-c]pyridine-4,7-dione (**2e**)

Yield: 28.9%; m.p. 65.7–66.3 °C; ^1^H NMR (600 MHz, methanol-*d*_4_) δ 7.84 (d, J = 8.4 Hz, 2H), 7.71 (d, *J* = 8.3 Hz, 2H), 7.52 (t, *J* = 7.6 Hz, 2H), 7.50–7.41 (m, 2H), 6.71–6.68 (m, 1H), 4.26 (t, *J* = 6.5 Hz, 1H), 4.06 (d, *J* = 7.0 Hz, 1H), 1.37–1.32 (m, 1H), 1.23 (d, *J* = 19.6 Hz, 3H), and 0.94 (t, *J* = 7.5 Hz, 6H). ^13^C NMR (150 MHz, methanol-*d*_4_) δ 175.14, 165.36, 146.84, 145.17, 123.29, 109.83, 58.33, 57.29, 49.57, 38.88, 38.31, 38.08, 27.34, 26.55, 16.04, 15.42, 11.98, and 11.62.

7-hydroxy-6-(sec-butyl)-furan[3,2-c]pyridin-4(5H)-one (**2f**)

Yield: 71.5%; m.p. 61.4–62.6 °C; ^1^H NMR (600 MHz, methanol-*d*_4_) δ 8.15 (s, 1H), 7.61 (d, *J* = 1.8 Hz, 1H), 6.89 (s, 1H), 4.64 (dd, *J* = 10.0, 4.9 Hz, 1H), 4.18 (qd, *J* = 6.4, 1.5 Hz, 2H), 3.57 (td, *J* = 6.6, 1.4 Hz, 2H), 1.77 (dddd, *J* = 23.5, 18.7, 11.6, 5.0 Hz, 3H), 1.70 (q, *J* = 6.6, 5.3 Hz, 2H), 1.56 (p, *J* = 6.7 Hz, 2H), 1.43 (q, *J* = 5.2, 3.7 Hz, 5H), 1.33 (s, 1H), 1.05–0.98 (m, 6H), and 0.96 (s, 2H).

5-(2- bromoethyl)-6-isobutyl-furano[3,2-c]pyridine-4,7-dione (**3a**)

Yield: 60%; m.p. 62.1–63.2 °C; ^1^H NMR (600 MHz, methanol-*d*_4_) δ 8.11 (dd, *J* = 1.6, 0.9 Hz, 1H), 7.57 (t, *J* = 1.8 Hz, 1H), 6.85 (dd, *J* = 2.0, 0.9 Hz, 1H), 4.65 (dd, *J* = 10.4, 4.3 Hz, 1H), 4.50–4.38 (m, 2H), 3.60 (t, *J* = 5.8 Hz, 2H), 1.82–1.70 (m, 3H), 1.30 (s, 1H), 1.03–0.97 (m, 4H), and 0.97–0.93 (m, 2H). ^13^C NMR (150 MHz, methanol-*d*_4_) δ 173.78, 165.44, 146.90, 145.24, 123.20, 109.73, 65.82, 52.36, 41.07, 29.70, 26.14, 23.27, and 21.80.

5-(3-bromopropyl)-6-isobutyl-furano[3,2-c]pyridine-4,7-dione (**3b**)

Yield: 37%; m.p. 47.8–49.1 °C; ^1^H NMR (600 MHz, methanol-*d*_4_) δ 8.11 (dd, *J* = 1.6, 0.9 Hz, 2H), 7.58 (t, *J* = 1.8 Hz, 2H), 6.85 (dd, *J* = 2.0, 0.9 Hz, 2H), 4.60 (dd, *J* = 10.0, 4.9 Hz, 2H), 4.28 (s, 2H), 4.26 (d, *J* = 6.0 Hz, 3H), 3.50 (t, *J* = 6.5 Hz, 5H), 2.18 (pd, *J* = 6.2, 4.2 Hz, 5H), 1.80–1.64 (m, 7H), 1.32–1.25 (m, 2H), 0.98 (dd, *J* = 22.4, 6.1 Hz, 15H), and 0.10 (s, 2H). ^13^C NMR (150 MHz, methanol-*d*_4_) δ 174.12, 165.45, 146.88, 145.28, 123.19, 109.69, 64.02, 52.45, 41.00, 32.78, 30.06, 26.16, 23.25, and 21.80.

5-(4-bromobutyl)-6-isobutyl-furano[3,2-c]pyridine-4,7-dione (**3c**)

Yield: 37%; m.p. 64.5–65.3 °C; ^1^H NMR (600 MHz, methanol-*d*_4_) δ 8.11 (dd, *J* = 1.6, 0.9 Hz, 2H), 7.58 (t, *J* = 1.8 Hz, 2H), 6.74 (dd, *J* = 2.0, 0.9 Hz, 2H), δ 4.39 (s, 1H), 4.11 (s, 1H), 4.09–4.03 (m, 1H), 3.49 (s, 1H), 3.52–3.45 (m, 0H), 3.39 (t, *J* = 7.0 Hz, 1H), 2.56 (s, 3H), 1.92–1.80 (m, 2H), 1.71 (d, *J* = 6.8 Hz, 1H), 1.62 (s, 3H), 1.53–1.46 (m, 1H), 1.30–1.11 (m, 3H), 0.91–0.84 (m, 1H), 0.87–0.76 (m, 1H), and 0.80 (s, 1H).

5-(5-bromopentyl)-6-isobutyl-furan[3,2-c]pyridine-4,7-dione (**3d**)

Yield: 52.5%; m.p. 56.2–57.1 °C; ^1^H NMR (600 MHz, methanol-*d*_4_) δ 8.00 (q, *J* = 4.5, 4.0 Hz, 1H), 7.46 (ddd, *J* = 8.7, 4.3, 2.2 Hz, 1H), 6.74 (td, *J* = 5.3, 2.8 Hz, 1H), 4.52–4.49 (m, 1H), 4.05–4.02 (m, 2H), 1.75 (dt, *J* = 7.3, 3.5 Hz, 2H), 1.60 (dd, *J* = 9.3, 5.3 Hz, 3H), 1.41 (ddd, *J* = 7.6, 4.9, 2.3 Hz, 2H), 0.93–0.85 (m, 9H), 0.84 (t, *J* = 5.1 Hz, 4H), and 0.79 (t, *J* = 5.8 Hz, 1H). ^13^C NMR (150 MHz, methanol-*d*_4_) δ 174.30, 165.37, 146.86, 145.24, 123.23, 109.71, 65.98, 52.45, 41.11, 34.11, 33.41, 28.80, 26.15, 25.62, 23.30, and 21.81.

5-(6-bromohexyl)-6-isobutyl-furano[3,2-c]pyridine-4,7-dione (**3e**)

Yield: 38.5%; m.p. 51.2–52.7 °C; ^1^H NMR (600 MHz, methanol-*d*_4_) δ 8.11 (dd, *J* = 1.6, 0.9 Hz, 1H), 7.58 (t, *J* = 1.8 Hz, 1H), 6.85 (dd, *J* = 1.9, 0.9 Hz, 1H), 4.60 (dd, *J* = 10.0, 4.8 Hz, 1H), 4.19–4.13 (m, 1H), 4.16–4.09 (m, 1H), 3.42 (t, *J* = 6.7 Hz, 2H), 1.88–1.61 (m, 6H), 1.51–1.25 (m, 5H), and 0.97 (dd, *J* = 21.8, 6.2 Hz, 6H). ^13^C NMR (150 MHz, methanol-*d*_4_) δ 174.34, 165.38, 146.86, 145.27, 123.25, 109.71, 66.12, 52.47, 41.09, 34.17, 33.83, 29.51, 28.71, 26.15, 23.29, and 21.80.

5,6-diisobutyl-furano[3,2-c]pyridine-4,7-dione (**3f**)

Yield: 29.6%; m.p. 48.2–49.1 °C; ^1^H NMR (600 MHz, methanol-*d*_4_) δ 8.12–8.09 (m, 1H), 7.57 (t, *J* = 1.8 Hz, 1H), 6.85 (dd, *J* = 2.0, 0.9 Hz, 1H), 4.62 (dd, *J* = 10.0, 4.8 Hz, 1H), 3.96–3.87 (m, 2H), 1.94 (dp, *J* = 13.3, 6.7 Hz, 1H), 1.80–1.70 (m, 2H), 1.72–1.64 (m, 1H), 1.30 (d, *J* = 10.9 Hz, 1H), 1.29 (s, 1H), and 1.25 (s, 3H), 1.01–0.88 (m, 15H). ^13^C NMR (150 MHz, methanol-*d*_4_) δ 174.17, 165.23, 146.68, 145.09, 123.12, 109.56, 72.08, 52.30, 48.87, 41.03, 28.86, 26.02, 23.14, 21.66, 19.18, and 19.15.

5-allyl-6-isobutyl-furano[3,2-c]pyridine-4,7-dione (**3g**)

Yield: 52.6%; m.p. 67.2–68.3 °C; ^1^H NMR (600 MHz, methanol-*d*_4_) δ 8.11 (t, *J* = 1.2 Hz, 1H), 7.62–7.55 (m, 1H), 6.85 (dd, *J* = 1.9, 0.9 Hz, 1H), 5.94 (ddt, *J* = 17.3, 10.8, 5.6 Hz, 1H), 5.33 (dq, *J* = 17.2, 1.7 Hz, 1H), 5.25–5.20 (m, 1H), 4.68–4.62 (m, 2H), 4.63 (dt, *J* = 2.7, 1.5 Hz, 1H), 1.81–1.65 (m, 3H), 1.32–1.25 (m, 1H), 1.07 (d, *J* = 6.6 Hz, 1H), 0.99 (d, *J* = 6.2 Hz, 3H), and 0.96 (d, *J* = 6.1 Hz, 3H).

5-ene-butyl-6-isobutyl-furano[3,2-c]pyridine-4,7-dione (**3h**)

Yield: 62.3%; m.p. 61.9~63.1 °C; ^1^H NMR (600 MHz, methanol-*d*_4_) δ 8.01 (dd, *J* = 1.6, 0.9 Hz, 2H), 7.51(t, *J* = 1.8 Hz, 1H), 6.78–6.73 (m, 2H), 5.70 (ddt, *J* = 17.1, 10.3, 6.8 Hz, 2H), 5.00 (dq, *J* = 17.2, 1.7 Hz, 2H), 4.94 (dq, *J* = 10.3, 1.4 Hz, 2H), 4.50 (dd, *J* = 10.0, 5.0 Hz, 2H), 4.14–4.00 (m, 5H), 2.30 (qt, *J* = 6.6, 1.4 Hz, 5H), 1.68–1.62 (m, 2H), 1.62 (d, *J* = 1.7 Hz, 1H), 1.62–1.56 (m, 2H), 1.58–1.51 (m, 1H), 1.25–1.15 (m, 1H), 0.96 (d, *J* = 6.7 Hz, 2H), and 0.86 (dd, *J* = 21.0, 6.1 Hz, 12H). ^13^C NMR (150 MHz, methanol-*d*_4_) δ 173.94, 165.09, 146.57, 144.95, 135.02, 122.96, 117.41, 109.43, 65.02, 52.09, 40.87, 33.87, 25.84, 22.99, and 21.50.

5-enopentyl-6-isobutyl-furan[3,2-c]pyridine-4,7-dione (**3i**)

Yield: 37.5%; m.p. 58.2~59.6 °C; ^1^H NMR (600 MHz, methanol-*d*_4_) δ 8.11 (dd, *J* = 1.6, 0.9 Hz, 1H), 7.57 (t, *J* = 1.8 Hz, 1H), 6.85 (dd, *J* = 2.0, 0.9 Hz, 1H), 5.81 (ddt, *J* = 17.0, 10.2, 6.7 Hz, 1H), 5.03 (q, *J* = 1.7 Hz, 1H), 5.02–4.96 (m, 1H), 4.98–4.92 (m, 1H), 4.61 (dd, *J* = 10.1, 4.9 Hz, 1H), 4.19–4.09 (m, 3H), 2.16–2.08 (m, 3H), 1.80–1.63 (m, 7H), 1.39 (d, *J* = 16.4 Hz, 0H), 1.32–1.25 (m, 2H), 1.07 (d, *J* = 6.6 Hz, 1H), 0.99 (d, *J* = 6.2 Hz, 4H), 0.96 (d, *J* = 6.2 Hz, 4H), and 0.10 (s, 2H). ^13^C NMR (150 MHz, methanol-*d*_4_) δ 174.30, 165.39, 146.86, 145.25, 138.71, 123.25, 115.77, 109.70, 65.63, 52.43, 41.11, 30.93, 28.93, 26.15, 23.28, and 21.86.

5-hexenyl-6-isobutyl-furan[3,2-c]pyridine-4,7-dione (**3j**)

Yield: 29.3%; m.p. 61.8–62.5 °C; ^1^H NMR (400 MHz, methanol-*d*_4_) δ 8.11 (dd, *J* = 1.3 Hz, 1H), 7.57 (t, *J* = 1.7 Hz, 1H), 6.90–6.82 (m, 1H), 5.78 (ddt, *J* = 17.0, 10.2, 6.7 Hz, 1H), 4.99 (dq, *J* = 17.2, 1.7 Hz, 2H), 4.93 (ddt, *J* = 10.3, 2.3, 1.2 Hz, 2H), 4.67–4.56 (m, 2H), 4.21–4.13 (m, 2H), 4.13–4.05 (m, 1H), 2.07 (dtt, *J* = 8.6, 6.9, 1.4 Hz, 3H), 1.81–1.69 (m, 3H), 1.69–1.59 (m, 4H), 1.59–1.38 (m, 3H), and 1.05–0.88 (m, 10H). ^13^C NMR (100 MHz, methanol-*d*_4_) δ 173.02, 145.55, 143.92, 138.19, 113.89, 108.37, 64.79, 51.12, 48.30, 48.09, 47.87, 47.66, 47.45, 47.24, 47.02, 39.76, 32.98, 27.77, 24.97, 24.80, 21.94, and 20.45.

5-ethylcyano-6-isobutyl-furan[3,2-c]pyridine-4,7-dione (**3k**)

Yield: 43%; m.p. 75.2–77.1 °C;^1^H NMR (600 MHz, methanol-*d*_4_) δ 8.10 (s, 1H), 7.56 (d, *J* = 2.2 Hz, 1H), 6.83 (s, 1H), 4.90 (d, *J* = 4.2 Hz, 2H), 4.87 (s, 3H), 4.67 (dd, *J* = 10.5, 4.4 Hz, 1H), 1.83–1.76 (m, 1H), 1.76–1.65 (m, 2H), 0.98 (d, *J* = 6.0 Hz, 3H), and 0.94 (d, *J* = 5.9 Hz, 3H). ^13^C NMR (150 MHz, methanol-*d*_4_) δ 172.99, 165.61, 147.15, 145.47, 123.15, 116.10, 109.81, 57.76, 57.61, 57.47, 52.13, 50.20, 40.83, 26.23, 23.36, and 21.84.

5-(6-(1H-imidazol-1-yl)hexyl)-6-isobutyl-5,6-dihydrofuro[3,2-c]pyridine-4,7-dione (**4a**)

Yield: 32%; m.p. 72.2–73.3 °C; ^1^H NMR (600 MHz, Methanol-*d*_4_) δ 8.14–8.09 (m, 2H), 7.66–7.61 (m, 2H), 7.58 (t, *J* = 1.8 Hz, 2H), 7.12 (s, 2H), 7.07 (s, 2H), 6.97 (s, 2H), 6.86 (d, *J* = 1.9 Hz, 2H), 4.60 (dd, *J* = 9.9, 5.0 Hz, 2H), 4.14 (t, *J* = 6.4 Hz, 4H), 4.01 (t, *J* = 7.1 Hz, 4H), 1.92 (s, 8H), 1.45–1.35 (m, 6H), 1.35–1.27 (m, 9H), and 0.98 (dd, *J* = 21.9, 6.2 Hz, 13H).

5-(6-(1H-benzo[d]imidazol-1-yl) hexyl)-6-isobutyl-5,6-dihydrofuro [3,2-c]pyridine-4,7-dione (**4b**)

Yield: 49.2%; m.p. 81.6–83.1 °C; ^1^H NMR (600 MHz, Methanol-*d*_4_) δ 8.04 (d, *J* = 7.9 Hz, 1H), 7.99 (dd, *J* = 1.6, 0.9 Hz, 1H), 7.57 (dt, *J* = 8.0, 1.0 Hz, 1H), 7.53–7.47 (m, 1H), 7.46–7.40 (m, 2H), 7.21 (ddd, *J* = 8.1, 7.1, 1.2 Hz, 1H), 7.19–7.12 (m, 2H), 6.72 (dd, *J* = 2.0, 0.9 Hz, 1H), 4.48 (dd, *J* = 10.0, 5.0 Hz, 1H), 4.15 (t, *J* = 7.2 Hz, 2H), 4.00 (t, *J* = 6.4 Hz, 2H), 1.74 (p, *J* = 7.3 Hz, 2H), 1.67–1.57 (m, 2H), 1.57–1.52 (m, 2H), 1.52–1.46 (m, 2H), 1.33–1.16 (m, 4H), and 0.84 (dd, *J* = 20.8, 6.3 Hz, 6H).

5-(6-(maleimido) hexyl hexyl)-6-isobutyl-5,6-dihydrofuro[3,2-c] pyridine-4,7-dione (**4c**)

Yield: 22.3%; m.p. 67.2–68.1 °C; ^1^H NMR (600 MHz, Methanol-*d*_4_) δ 8.11 (s, 2H), 7.57 (d, *J* = 1.8 Hz, 1H), 6.85 (d, *J* = 1.1 Hz, 1H), 4.64 (t, 1H), 4.20 (t, 2H), 3.05 (t, *J* = 7.3 Hz, 2H), 1.76–1.61 (m, 4H), 1.61–1.45 (m, 3H), 1.43–1.21 (m, 6H), and 0.97 (dd, *J* = 14.2, 6.3 Hz, 6H).

^13^C NMR (150 MHz, Methanol-*d*_4_) δ 181.82, 180.30, 174.57, 145.50, 165.02, 153.50, 143.12, 123.91, 106.43, 60.09, 44.80, 38.15, 37.62, 29.27, 28.03, 27.74, 27.44, 27.11, 26.81, 25.51, 23.50, and 22.00.

5-(6-(succinimide)hexyl)-6-isobutyl-furo[3,2-c]pyridine-4,7-dione (**4d**)

Yield: 38.7%; m.p. 70.1–71.3 °C; ^1^H NMR (400 MHz, Methanol-*d*_4_) δ 8.14–8.08 (m, 1H), 7.57 (t, *J* = 1.8 Hz, 1H), 6.85 (dd, *J* = 2.0, 0.8 Hz, 1H), 4.13 (d, *J* = 4.2 Hz, 2H), 3.45 (t, *J* = 7.3 Hz, 3H), 2.69 (d, *J* = 2.6 Hz, 9H), 1.93 (s, 2H), 1.72–1.60 (m, 5H), 1.54 (d, *J* = 7.3 Hz, 3H), 1.36–1.26 (m, 6H), and 1.02–0.92 (m, 8H). ^13^C NMR (100 MHz, Methanol-*d*_4_) δ 181.60, 180.09, 174.35, 146.90, 145.28, 123.24, 109.71, 66.10, 52.47, 49.64, 49.42, 49.28, 49.21, 49.00, 48.79, 48.57, 48.36, 41.07, 39.39, 30.57, 29.48, 29.06, 28.50, 27.34, 26.46, 26.14, 23.29, and 21.78.

## 3. Materials and Methods

### 3.1. Materials

All solvents and reagents used in this study were obtained from commercial suppliers by McLean, Titan Technologies Inc and were used directly without further purification. Melting points were measured using a WRX-4 micromelting point meter. The microwave reactor model used was a WBFY-205 (Shanghai Jingke Industrial Co., Ltd., Shanghai, China). The column chromatography was performed over pure silica gel 60 (200–300 mesh, Qingdao Chemical Co., Qingdao, China). Thin-layer chromatography (TLC) was performed on silica gel 60 GF254 (Qingdao Hai Yang Chemical Co., Ltd., Qingdao, China). ^1^H and ^13^C NMR spectra were recorded in CH_3_OD or CDCl_3_ as the solvent using an AVANCE III HD 600 MHz spectrometer (Bruker Co., Ltd., Fällanden, Switzerland).

### 3.2. Synthesis

#### 3.2.1. N-(3-Furanoyl)-L-leucine (**2a**)

An amount of 2.00 g (17.79 mmol, 1.0 eq.) of 3-furan formic acid was added, which was dissolved in 4 mL of anhydrous dichlorosulfoxide (53.37 mmol, 3.0 eq.). Later, the reaction was set to reflux for four hours at 70 °C, followed by evaporation of the solvent to obtain the newly prepared 3-furancarboxylic acid chloride.

An amount of 2.00 g (15.22 mmol, 1.0 eq.) of L-leucine was added, which was dissolved in 15 mL of anhydrous dichloromethane. After complete dissolution, the solution was stirred under ice bath conditions for 5 min. Then, the prepared 3-furanoyl chloride was added by a rubber head dropper. Once the addition was completed, the reaction was stirred at 25 °C for 1 h. After the reaction was completed, quenched by water, and the pH was adjusted to 3–4. Trichloromethane was used as an extractant 3 times, the organic phases were merged, and column chromatography separation was performed using a silica gel column (V_PE_:V_EA_ = 5:1). N-(3-furanoyl)-L-leucine (**2a**) was obtained.

#### 3.2.2. 6-Isobutyl-furan [3,2-c] pyridin-4,7-dione (**2b**)

An amount of 0.50 g (2.22 mmol, 1.0 eq.) of N-(3-furanoyl)-L-leucine (**2a**) was added, which was dissolved in 5 mL of Eaton reagent and stirred at 110 °C for 5 h. After the reaction was completed, an ice water mixture was added to quench the reaction, the reaction solution was filtered, and the pH was adjusted to neutral. The solution was extracted three times with ethyl acetate and column chromatography separation was performed (V_PE_:V_EA_ = 2:1) on a silica gel column to obtain the furano–nitrogen heterocyclic compound 6-isobutyl-furan [3,2-c] pyridin-4,7-dione (**2b**).

#### 3.2.3. 6-Isobutyl-furan [3,2-c] pyridin-4 (5H)-ketone (**2c**)

An amount of 0.30 g (1.45 mmol, 1.0 eq.) of 6-isobutyl-furan [3,2-c] pyridine-4,7-dione (**2b**) was dissolved in 5 mL of anhydrous ethanol. Subsequently, 0.27 g of NaBH_4_ (7.24 mmol, 5.0 eq.) was added slowly to the solution with stirring to ensure controlled reactivity. The reaction mixture was then stirred at room temperature for a period of 1 h. After the reaction was completed, an ice water mixture was added to quench the reaction, the reaction solution was filtered and extracted three times with ethyl acetate, and column chromatography separation was performed using a silica gel column (V_PE_:V_EA_ = 6:1) to obtain 7-hydroxy-6-isobutyl-furan [3,2-c] pyridin-4 (5H)-ketone (**2c**).

#### 3.2.4. 5-(2-Bromoethyl)-6-isobutyl-furano[3,2-c]pyridine-4,7-dione (**3a**)

An amount of 0.30 g (1.45 mmol, 1.0 eq.) of 6-isobutyl-furan [3,2-c] pyridine-4,7-dione (**2b**) was dissolved in 5 mL of anhydrous acetonitrile and 0.60 g of anhydrous potassium carbonate (4.34 mmol, 3.0 eq.), and 0.16 mL of dibromoethane (1.88 mmol, 1.3 eq.) was added. The solution was stirred and refluxed at 70 °C for 4 h. Following this, insoluble substances were filtered out, the solution was extracted with saturated sodium bicarbonate solution and ethyl acetate, the organic phase was collected, and the solvent was removed by rotary evaporation. White powder of 5-(2-bromoethyl)-6-isobutyl-furan [3,2-c] pyridine-4,7-dione (**3a**) was obtained by column chromatography separation (V_PE_:V_EA_ = 8:1) with a silica gel column.

#### 3.2.5. 5-(6-(1H-Imidazol-1-yl)hexyl)-6-isobutyl-5,6-dihydrofuro[3,2-c]pyridine-4,7-dione (**4a**)

An amount of 0.30 g (0.81 mmol, 1.0 eq.) of 5-(6-bromohexyl)-6-isobutyl-furan [3,2-c] pyridine-4,7-dione (**3e**) was dissolved in 5 mL of anhydrous acetonitrile, and 0.34 g of anhydrous potassium carbonate (2.43 mmol, 3.0 eq.) and 0.07 g of imidazole (1.05 mmol, 1.3 eq.) were added to the reaction, stirred, and refluxed at 70 °C for four hours. Following this, insoluble substances were filtered out, the solution was extracted with saturated sodium bicarbonate solution and ethyl acetate, the organic phase was collected, and the solvent was removed by rotary evaporation; a silica gel column was used for column chromatography separation (V_PE_:V_EA_ = 3:1) to obtain a white powder of 5-(6-(1H-imidazole-1-yl) hexyl)-6-isobutyl-furan [3,2-c] pyridine-4,7-dione (**4a**).

### 3.3. Cancer Cell Viability Assay

Novel furan pyridine derivatives were screened for their antitumor activities on two cancer cell lines (KYSE70 and KYSE150) following the MTT method.

Cancer cell lines, including two human esophageal cancer cells KYSE70 and KYSE150, provided by the Institute of Procell Life Science Technology Co., Ltd. (Wuhan, China).

In vitro cytotoxicity studies were conducted on KYSE-70 cells and KYSE-150 cells using the MTT assay. The MTT assay was performed following a common protocol. Simply put, cells were seeded on a 96-well plate (50 µL, 4 × 10^4^ cells/well). After 4 h, 50 µL of fresh medium containing CB01 was added to each well at the indicated concentration. After 48 h, 0.1 mg/mL of MTT solution was added to each well and incubated for 4 h. After removing the supernatant, 200 µL of 100% (*w*/*v*) DMSO was added to each well and incubated at room temperature (25 °C) for 10 min. Finally, the absorbance of the sample was measured at 595 nm using an enzyme-linked immunosorbent assay reader. The experiment was repeated three times, each in triplicate [22].

### 3.4. Molecular Docking Simulations

Molecular docking studies offer important information with regard to molecule–protein interactions, prediction of binding configuration for each selected molecule at the active sites of target protein, and binding energies (binding affinity), and provide an efficient way to design more potent inhibitors [23,24]. MetAP2 is highly expressed in tumor cells (such as esophageal cancer) and plays an important role in tumor growth and metastasis [25]. Inhibiting MetAP2 is considered to be an important pathway for inhibiting angiogenesis during the growth and metastasis of solid tumors [26]. Therefore, MetAP2 inhibitors have great potential in the design and development of anti-angiogenic drugs [27].

The compounds with more excellent activity were investigated via docking methodology against METAP2 and EGFR proteins. The X-ray crystal structure of two proteins was obtained from the Protein Data Bank (https://www.rcsb.org/, accessed on 20 April 2024). Molecular docking of compounds with excellent biological activity with three-dimensional X-ray structures of two proteins was carried out using a molecular manipulation environment (MOE 2019). Compound structures were built using the builder interface of the MOE program and subjected to energy minimization using the included Forceeld MMFF94x calculations.

## 4. Conclusions

A series of furan[3,2-c] pyridine derivatives were designed and synthesized based on the principle of active substructure splicing. The anti-tumor activity investigation revealed that the prepared compounds showed significant in vitro anti-tumor activities. Among them, compound **4c** expressed strong anti-tumor activity, which demonstrated an exceptional level of cytotoxicity against KYSE70 and KYSE150 esophageal cancer cell lines, achieving a 99% inhibition rate of cell growth. After administration at a concentration of 20.00 µg/mL, IC_50_ was 0.888 µg/mL within 24 h and 0.655 µg/mL within 48 h. Molecular docking of the active compound **4c** with the EGFR (PDB ID: 6DUK) and MetAP2 (PDB ID: 5D6E) proteins, which are potential targets in esophageal cancer, revealed hydrogen bonding interactions between the functional groups of the compound, including the furan ring and carbonyl groups, with specific amino acid residues. Accordingly, compound **4c** was identified as the most promising candidate in this work. This study discovered a series of furapyridone derivatives with significant anti-tumor activities and provided guidance to develop novel anti-tumor agents from furan-fused N heterocycle.

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
