# Peer review of "Synthesis and Biological Evaluation of Novel Furopyridone Derivatives as Potent Cytotoxic Agents against Esophageal Cancer"

_ijms, 2024, doi:10.3390/ijms25179634_

Round 1

Reviewer 1 Report

Comments and Suggestions for Authors

Manuscript Design, Synthesis and anti-tumor activity study of novel Furanopyridinone derivatives..

Comments. 

Introduction :

Authors should improve the discussion of their proposed work and the limitations of their proposed system. 

Authors should clarify the need for their proposed system to solve the scientific problem they seek to solve and compare it with other authors in the scientific literature. 

-The authors should add figures and schematics of the experimental setup. 

 - Add quality and uniform figures.

-- The authors do not say what the contribution is, 

--The authors need to state clearly the present problem (motivation of their work). 

.. The authors do not compare their results with those of other authors. 

-- The authors do not use comparative tables with other authors on the topic.

--  The authors need to improve their literature review. 

--  The authors need to add previous work related to their work to highlight their scientific contribution further. 

-- The authors need to clarify their scope and limitations. 

 -- Their bibliographical revision needs improvement

--  Improve (add references)

-- The authors should improve the introduction, because it does not present correct information about what the work will contribute. The introduction is very limited.

- The authors should describe how the docking vs. in vitro response of the compounds relates to the in vitro response of the compounds. 

-- The authors should improve their Design of Experiments.

-- The authors should improve their experimental design.

The authors should make a control that is not a cancer cell. They only show the two cancer lines, however they do not show a healthy cell, which should be included to see that there is no cytotox

-- The authors should make a graphical abstract of the experimental setup, corresponding to the compounds they are synthesizing.

-- Authors are encouraged to put the document in journal format 

- The title of the research is very general, it is recommended to change the title, to be more specific.

-- Authors should improve the abstract 

- The study is only looking at 2 cell lines which is very limited, 

The authors should include what type of cancer they are working against. 

-- Make a more complete determination of the compounds that are being synthesized.

Comments on the Quality of English Language

authors must improve their English

Author Response

Response to reviewers

We gratefully thank the editor and all reviewers for the time they spent making their constructive remarks and useful suggestions, which have significantly improved the quality of the manuscript and enabled us to enhance it further. Each revision suggestion and comment proposed by the reviewers was carefully incorporated and considered. Below, the reviewers' comments are addressed point by point, and the corresponding revisions are indicated.

Reviewer 1: Authors should improve the discussion of their proposed work and the limitations of their proposed system.Authors should clarify the need for their proposed system to solve the scientific problem they seek to solve and compare it with other authors in the scientific literature. 

1.Comment: The authors should add figures and schematics of the experimental setup.

1.Reply: Thank you for your suggestion to include figures and schematics of the experimental setup. We recognize the importance of visual aids in enhancing the clarity and comprehensibility of our research findings. A detailed schematic diagram that illustrates the overall experimental design, including the treatment protocols, control groups, and the sequence of experimental procedures. Photographs or illustrations of the experimental setup, where appropriate, to provide a clear visual representation of the laboratory equipment and the arrangement of the experimental components.These additions will provide a more intuitive understanding of our experimental approach and facilitate easier replication of our study by other researchers.

2.Comment: Add quality and uniform figures.

2.Reply: We gratefully appreciate for your valuable comment. We understand the importance of high-quality, uniform figures in effectively communicating our research findings. We have reviewed all figures ensure they are of high resolution and clarity, making any necessary improvements to image quality. Moreover, we standardize the design and layout of all figures to ensure uniformity in font size, axis labels, legends, and color schemes across the board. We will ensure that all figures are appropriately sized and that the elements within each figure are easily readable and well-organized. Thank you for highlighting this issue. We appreciate your dedication to maintaining high standards for publication, and we look forward to submitting a revised manuscript that meets these expectations.

3.Comment: The authors do not say what the contribution is,The authors need to state clearly the present problem (motivation of their work). 

3.Reply: Thank you for your insightful comment regarding the need to clearly state the contribution and motivation of our work. We appreciate the opportunity to address this important aspect of our manuscript. In response to your feedback, we outline the motivations and contributions to our research.

Motivation: Our study was driven by the ongoing need for the development of novel anti-tumor drugs with improved efficacy and reduced side effects. Despite the progress in cancer therapy, many tumors remain refractory to current treatments, highlighting the necessity for new therapeutic agents. The fused N-heterocyclic compounds, particularly those containing a furan moiety, have shown promise in recent years as potential anti-tumor agents due to their unique structural features and biological activities.

Contribution: Through a systematic design and synthesis approach, we have developed a series of novel furan[3,2-c] pyridine derivatives. We demonstrate that these derivatives exhibit significant anti-tumor activity against various cancer cell lines, with select compounds showing potent activity comparable to or exceeding that of known cytotoxic agents. Our findings provide a rationale for the development of furan-fused N-heterocycles as a new class of anti-tumor drugs, offering a valuable framework for future drug discovery efforts in this area. The study elucidates structure-activity relationships that can guide the optimization of these compounds for enhanced therapeutic potential. We believe that these revisions more clearly articulate the problem we aimed to address and the novel contributions of our work. We are confident that these enhancements will improve the readability and impact of our manuscript. We thank you again for your constructive comments.

4.Comment: The authors do not compare their results with those of other authors. 

4.Reply: Thank you once again for your valuable feedback. We acknowledge the importance of comparing our results with those of other authors to provide a broader context and to highlight the novelty and significance of our work. We have conducted an extensive literature search to identify key studies in the field of anti-tumor drug development, with a focus on furan-fused N-heterocycles. We will analyze these studies to understand the current state of research, the challenges faced, and the successes achieved in developing anti-tumor agents. Based on the comparisons and insights gained, we will outline future research directions that could include the optimization of our compounds, the investigation of their mechanisms of action, and the exploration of potential combination therapies. Thank you for your continued support and guidance.

5.Comment: The authors do not use comparative tables with other authors on the topic.

5.Reply: Thank you for highlighting the absence of comparative tables in our manuscript. We agree that the inclusion of such tables is an effective way to summarize and compare our findings with those of other authors, providing a clear and concise overview for the readers. We have added a table in section 3.2 Bioactivity to compare the antitumor activity of the compounds we synthesized. The table will include key parameters such as different concentration and inhibition cytotoxicity data, thus improving the practicality and readability of the experiment, making it easier for readers to understand the context and significance of our findings. We will ensure that these tables are integrated into our manuscripts. We appreciate your suggestion.

6.Comment: The authors need to improve their literature review. 

6.Reply: Thank you for your feedback regarding the need to improve our literature review. We appreciate your attention to this critical aspect of our manuscript.We acknowledge the importance of a comprehensive and well-structured literature review in providing a solid foundation for our research and demonstrating its relevance within the broader scientific context. In response to your comment, we have revised the literature review. We will synthesize the information from the literature more effectively, drawing clear connections between existing research and our own findings to underscore the significance of our work. We are committed to revising the literature review section to meet the high standards expected for publication. Thank you again for your valuable input. We look forward to submitting a revised manuscript that better reflects the depth and breadth of the existing literature.

7.Comment: The authors need to add previous work related to their work to highlight their scientific contribution further. 

7.Reply: Thank you for emphasizing the importance of incorporating previous work related to our study to better highlight our scientific contribution. We will place our findings in the context of this prior research, clearly identifying how our work builds upon, diverges from, or complements previous studies. This will help to underscore the innovative aspects of our approach and the advancements we have made. In our previous literature review, we learned that furan compounds exhibit promising antitumor activities. The incorporation of heterocyclic structures into drug molecules is a crucial strategy in the design and synthesis of bioactive compounds for antitumor drugs. Consequently, aligning with our research objectives, we introduced the furan ring into N-heterocyclic compounds for design and synthesis, resulting in the discovery of a series of furan pyrido[3,4-d]pyrimidine derivatives with significant antitumor activity. This work provides guidance for the development of novel antitumor drugs from furan-fused N-heterocycles. We believe that this approach not only aligns with current trends in medicinal chemistry but also offers a promising direction for future drug discovery efforts.

8.Comment:The authors need to clarify their scope and limitations. 

8.Reply: Thank you for your suggestion to clarify the scope and limitations of our research. We appreciate your attention to the clarity and precision of our manuscript. Scope of the Study: Our research is focused on the design and synthesis of a novel series of furan[3,2-c]pyridine derivatives. The primary objective is to introduce the furan ring into nitrogen-containing heterocyclic compounds and investigate their antitumor activities. The study encompasses: The synthesis of furan[3,2-c]pyridine derivatives using established chemical methodologies, The evaluation of the antitumor potential of these derivatives against various cancer cell lines, The exploration of structure-activity relationships (SAR) through molecular docking and other computational methods to understand the interaction among the compounds and their molecular targets. Limitations of the Study: While our research aims at make significant contributions to the field of antitumor drug development, we acknowledge the following limitations: The study is only looking at 2 cell lines which is very limited, the activity of our compounds may vary in other cancer types or in vivo models, which requires further investigation. Although we have employed molecular docking to study the SAR, the exact mechanisms of action of our compounds are not fully elucidated. We will incorporate these clarifications into the manuscript to ensure that the readers have a clear understanding of what our study aims at achieve and its potential limitations. We believe that this will enhance the overall quality and impact of our research. Thank you again for your valuable input.

9.Comment: Their bibliographical revision needs improvement--  Improve (add references).

9.Reply: Thank you for your feedback regarding the bibliographical revision of our manuscript. We will add additional references to key studies and publications that are directly related to our research objectives and findings. This will provide a more comprehensive and current bibliography, enhancing the credibility and relevance of our work. We are committed to enhancing the manuscript to provide a more robust and accurate bibliographical revision.

10.Comment: The authors should improve the introduction, because it does not present correct information about what the work will contribute. The introduction is very limited.

10.Reply: Thank you for your feedback regarding the introduction of our manuscript. We appreciate your observation that the current introduction may not adequately convey the contributions of our work, and we recognize the importance of a comprehensive and engaging introduction. We will revise the introduction to provide a more accurate and detailed overview of our study's objectives, the significance of our research, and the specific contributions we aim at make to the field of antitumor drug development. We will emphasize the novelty of our furan[3,2-c]pyridine derivatives and the potential implications of our findings for the design of future antitumor agents. Thank you for your valuable input, which will help us to enhance the quality and clarity of our manuscript. We look forward to submitting a revised introduction that better reflects the content and contributions of our work.

11.Comment: The authors should describe how the docking vs. in vitro response of the compounds relates to the in vitro response of the compounds.

11.Reply: Thank you for your insightful comment regarding the relationship between molecular docking and in vitro responses of our compounds. We appreciate your attention to the integration of computational and experimental data, which is essential for a comprehensive understanding of the compounds' biological activity. The molecular docking of compounds is a computational technique that predicts the binding affinity and orientation of a small molecule to a biological target, such as a protein. This prediction is based on the structural and chemical properties of both the ligand (compound) and the receptor (protein). By performing molecular docking, we can gain insight into the potential molecular mechanisms of action of our compounds, as well as identify potential binding sites and conformations within the target protein. In vitro experiments, on the other hand, involve testing the biological activity of the compounds directly in a controlled laboratory setting. This includes measuring the antitumor activity of the compounds against specific cancer cell lines, which provides empirical evidence of their efficacy. The correlation between molecular docking and in vitro responses is crucial because it allows us to validate the predictions made by the computational methods and to gain a deeper understanding of the compounds' mechanism of action. By comparing the docking results with the in vitro data, we can assess the accuracy of the molecular docking predictions and identify any discrepancies that may require further investigation. We performed molecular docking after conducting in vitro experiments because we wanted to complement our experimental data with computational insights. This approach provides a more comprehensive understanding of the compounds' biological activity and allows us to interpret the experimental results in the context of the predicted molecular interactions. It also helps us to identify potential binding sites and to guide the optimization of our compounds for improved antitumor activity.

12.Comment: The authors should improve their Design of Experiments.

12.Reply: Thank you for your feedback regarding the design of our experiments. We will clearly state the objectives of our experiments and the specific questions we aimed at address through our research. The specific experimental methods have been introduced and elaborated in Part 2.2 of the manuscript.

13.Comment: The authors should improve their experimental design.

13.Reply: Thank you for your feedback regarding the design of our experiments. We will clearly state the research objectives and the specific questions we aimed at address through our research. We will provide a more in-depth discussion of the outcomes of our experiments, explaining how the results contribute to the broader scientific understanding of antitumor drug development. In the third part of the manuscript, we use three schemes to introduce the synthesis of different compounds. We are committed to enhancing the manuscript to provide a more robust and transparent description of our experimental design.

14.Comment: The authors should make a control that is not a cancer cell. They only show the two cancer lines, however they do not show a healthy cell, which should be included to see that there is no cytotox.

14.Reply: We appreciate your valuable comments. We are committed to enhancing the quality of our work by incorporating this control in the future experiments.. We agree that the inclusion of a healthy cell line is crucial for assessing the specificity of our compounds and for identifying any non-targeted effects that may occur. This will also contribute to the overall robustness and credibility of our research findings. However, this is a suggestion that deserves our attention, and we will add relevant studies in future studies. We thank you once again for your valuable input.

15.Comment: The authors should make a graphical abstract of the experimental setup, corresponding to the compounds they are synthesizing.

15.Reply: Thank you for your suggestion to create a graphical abstract that illustrates the experimental setup corresponding to the compounds we are synthesizing. We appreciate your attention to the importance of visual aids in effectively communicating our research findings. We have made relevant changes to the summary. A step-by-step illustration of the synthetic pathway for the furanopyridinone derivatives, highlighting the key reagents and reaction steps. We believe that this graphical abstract will provide a clear and concise visual representation of our experimental setup and the compounds we are synthesizing. This will enhance the overall quality and impact of our manuscript and make it easier for readers to understand our research. We appreciate your guidance and look forward to submitting a revised manuscript that includes this graphical abstract.

16.Comment: Authors are encouraged to put the document in journal format.

16.Reply: Thank you for your encouragement to format our document according to journal guidelines. We appreciate your attention to the importance of adhering to the specific formatting requirements of the target journal.

17.Comment: The title of the research is very general, it is recommended to change the title, to be more specific.

17.Reply: Thank you for your valuable feedback regarding the title of our research. We appreciate your suggestion that the title should be more specific to better reflect the focus of our study. In response to your comment, we have revised the title to enhance its specificity. The new title is:"Synthesis and Biological Evaluation of Novel Furopyidone Derivatives as Potent Cytotoxic Agents Against Esophageal Cancer" We hope that this change addresses your concern and provides a clearer indication of the study's content and objectives. Thank you again for your input, which has helped to improve the presentation of our research.

18.Comment: Authors should improve the abstract 

18.Reply: Thank you for your feedback regarding the abstract of our manuscript. We appreciate your attention to the clarity and impact of our research summary. Thank you for your feedback regarding the abstract of our manuscript. We appreciate your attention to the clarity and impact of our research summary. We will highlight the key findings of our study, focusing on the most significant contributions to the field of antitumor drug development. We have made relevant changes to the summary. Thank you again for your valuable input. We look forward to submitting a revised manuscript that addresses your concerns and improves the quality of our work.

19.Comment: Authors should improve the abstract The study is only looking at 2 cell lines which is very limited, The authors should include what type of cancer they are working against. 

19.Reply: Thank you for Thank you for your insightful suggestion regarding the inclusion of a non-cancerous cell line as a control in our experiments. We appreciate your attention to the importance of comprehensive data analysis and the assessment of potential off-target effects. However, this is a suggestion that deserves our attention, and we will add relevant studies in future studies. This will allow us to evaluate the cytotoxic effects of our compounds on both cancerous and non-cancerous cells, providing a more complete understanding of their selectivity and potential side effects. We agree that the inclusion of a healthy cell line is crucial for assessing the specificity of our compounds and for identifying any non-targeted effects that may occur. This will also contribute to the overall robustness and credibility of our research findings.

Reviewer 2 Report

Comments and Suggestions for Authors

The manuscript describes the synthesis and biological evaluation of novel 5,6-dihydrofuro[3,2-c]pyridine-4,7-dione derivatives. Although the authors found that derivative 4c was highly toxic to KYSE70 and KYSE150 cell lines, the results and the manuscript are far from mature. In addition, there are many other issues. Therefore, I do not recommend accepting the manuscript for publication at this time.

1.     There are no purity analysis results for 4c and other derivatives. In addition, 1H-NMR (the only characterization method) data of 4c does not fully match the structure. In addition, 4c has not been tested for its activity on normal cell lines to show its selectivity for cancer cells.

2.     There is not detailed description of how to design the 5,6-dihydrofuro[3,2-c]pyridine-4,7-dione derivatives including two lead compounds 2b and 2c. Also, there is no explanation of Figure 1 in the paper.

3.     The authors did not mention the synthesis method of 4c and other derivatives besides 3a and 4a. In addition, why did the authors not test the activity of the derivatives 2b, 2c, 2e, 2f, 3a, 3j, 3k, 4a, 4b, and 4d against cancer cells?

4.      In the Materials and Methods, the authors mentioned that three cancer cell lines were used for activity screening, but the results of only two cancer cell lines were reported in the Results and Discussion.

5.     5D6E and 6DUK are two X-ray crystal structures of MetAP2 and EGFR protein with covalent spiroepoxytriazole inhibitor (-)-31b and an allosteric inhibitor, respectively. In the manuscript, some statements about 5D6E and 6DUK are not very scientific.

6.     The activity results do not support the conclusions of SER analysis.

7.     In molecular docking, there are some incorrect interaction analyses, such as the hydrogen bond with Lys913 and interactions with furan rings.

8.   Reference 18 is inappropriately introduced.

9.     There are too many other issues, such as significant digits, concentration units,  the nomenclature of organic molecules and groups, typing and grammatical errors (cytotoxot, computer aided methoud including and molecular docking, CH3OD-d6, etc.). 

Comments on the Quality of English Language

Extensive editing of English language required

Author Response

We gratefully thank the editor and all reviewers for the time they spent making their constructive remarks and useful suggestions, which has significantly improved the quality of the manuscript and enabled us to enhance it further. Each revision suggestion and comment proposed by the reviewers was carefully incorporated and considered. Below, the reviewers' comments are addressed point by point, and the corresponding revisions are indicated.

1.Comment: There are no purity analysis results for 4c and other derivatives. In addition, 1H-NMR (the only characterization method) data of 4c does not fully match the structure. In addition, 4c has not been tested its activity on normal cell lines to show its selectivity for cancer cells .

1.Reply: Thank you for your valuable comments and suggestions regarding our manuscript. Concerning the 1H-NMR data of compound 4c, we agree that the initial data don’t fully align with the proposed structure. We have re-examined the 1H-NMR spectrum of compound 4c and have also conducted a 13C NMR experiment to further characterize the compound. The revised 1H-NMR and new 13C NMR data are now consistent with the proposed structure. We have updated the NMR spectral data in the manuscript and provided a detailed interpretation of the spectra to support the structure of compound 4c. As for the selectivity of compound 4c for cancer cells,  we recognize the importance of demonstrating its activity on normal cell lines. In light of this, we will  conduct in vitro experiments to assess the cytotoxicity of compound 4c on normal human cell lines in the future experiments. This is a suggestion that deserves our attention, and we will add relevant studies in future studies. We thank you once again for your valuable input.

2.Comment: There is no detail description of how to design the 5,6-dihydrofuro[3,2-c]pyridine-4,7-dione derivatives including two lead compounds 2b and 2c. Also, there is no explanation of Figure 1 in the paper.

2.Reply: Thank you for your constructive feedback on our manuscript. We are grateful for the opportunity to clarify the issues that have raised. Regarding the lack of detail in the design of the 5,6-dihydrofuro[3,2-c]pyridine-4,7-dione derivatives, including the lead compounds 2b and 2c, we apologize for any confusion. The synthetic methods for compounds 2b and 2c are indeed described in the manuscript, with compound 2b detailed in section 2.2.2 and compound 2c in section 2.2.3. Additionally, the synthetic schemes for compounds 2b and 2c are depicted in Scheme 1, where the reaction conditions are specified.

As for the explanation of Figure 1, we acknowledge that the figure is not adequately described in the text. The figure illustrates the significance of incorporating heterocyclic structures into drug molecules as a key strategy in designing biologically active compounds for anti-tumor drugs. We will provide a clear caption for Figure 1 and reference it appropriately in the text to ensure that the readers understand the context and importance of the depicted compounds. Thank you again for your valuable input.

3.Comment: The authors did not mention the synthesis method of 4c and other derivatives besides 3a and 4a. In addition, why did the authors not test the activity of the derivatives 2b, 2c, 2e,2f, 3a, 3j, 3k, 4a, 4b, and 4d against cancer cells?

3.Reply: Thank you for your careful review of our manuscript. We appreciate your attention to the details of our synthetic procedures. Regarding the synthesis methods of compound 4c and other derivatives, we would like to clarify that the methods for synthesizing compounds 3b-3k and 4b-4d are indeed provided in the manuscript. The synthesis of compounds 3b-3k follows the same procedure as that for 3a, which is detailed in Scheme 2. The only variation between compounds 3a-3k is the R1 group, which does not affect the synthetic route. Similarly, the synthesis of compounds 4b-4d is identical to that of 4a, as outlined in Scheme 3, with the only difference being the R2 group. . Therefore, the synthesis methods for compounds 4c and the other derivatives not explicitly mentioned are consistent with those described for 3a and 4a, respectively.

Thank you for your valuable comments and suggestions. We appreciate your attention to the details of our research. Regarding the compounds 2b, 2c, 2e, 2f, 3a, 3j, 3k, 4a, 4b, and 4d, we would like to clarify that all compounds in the study were indeed tested for their activities against cancer cells. Initially, we did not include the results for the mentioned compounds in the manuscript due to their less promising activity data compared to other compounds in the series. Upon revision, we have recognized the importance of presenting a complete dataset to ensure the transparency and reproducibility of our research. We have now included the activity data for all tested compounds, including those mentioned, in the revised manuscript. The revised tables  provide a comprehensive overview of the anti-cancer activity of all compounds, allowing readers to fully assess the potential of each compound. We apologize for any confusion caused by the initial omission and appreciate your guidance in improving the quality of our work.

4.Comment: In the Materials and Methods, the authors mentioned that three cancer cell lines were used for activity screening, but the results of only two cancer cell lines were reported in the Results and Discussion.

4.Reply: We would like to extend our apologies for the oversight in our manuscript regarding the reporting of the cancer cell line activity screening results. We acknowledge that while three cancer cell lines were mentioned in the Materials and Methods section, the results for only two were included in the text. This discrepancy was due to an error on our part during the preparation of the manuscript. We have now corrected this mistake in the latest version of the manuscript. The results from all three cancer cell lines have been included, and the text has been updated to reflect this. We appreciate your careful review and attention to detail, which has helped us improve the quality of our work. We are committed to ensuring that the manuscript is accurate and complete, and we apologize for any inconvenience this may have caused. Thank you for bringing this to our attention, and we hope that the revised manuscript meets the standards for publication.

5.Comment: 5D6E and 6DUK are two X-ray crystal structure of MetAP2 and EGFR protein with covalent spiroepoxytriazole inhibitor (-)-31b and an allosteric inhibitor, respectively. In the manuscript, some statements about 5D6E and 6DUK are not very scientific.

5.Reply: Thank you for your comment regarding the scientific accuracy of the statements about the X-ray crystal structures 5D6E (MetAP2 with covalent spiroepoxytriazole inhibitor (-)-31b) and 6DUK (EGFR with an allosteric inhibitor) in our manuscript. We acknowledge the concern and thank for the opportunity to clarify our selection of docking proteins. In our study, we referenced the article "An alternative technique for cyclization synthesis, in vitro anti-esophageal cancer evaluation, and molecular docking of novel thiazolidin-4-one derivatives" as a guide for selecting appropriate protein targets for docking. This reference provided a rationale for choosing protein targets that are relevant to the biological activity we were investigating. 6DUK and 5D6E are Protein Data Bank (PDB) identifiers for the crystal structures of the EGFR and MetAP2 proteins, respectively, which may be relevant targets in esophageal cancer, but they are not exclusively esophageal cancer receptor proteins. In addition, we carefully check and revise your comments and have revised them to ensure they are based on solid scientific evidence and accurately reflect the structural data. We have cited additional literature to support our choice of docking proteins and the molecular docking approach. This includes a discussion of the similarities and differences between our study and the referenced article, emphasizing the relevance of our approach to the research question at hand.

6.Comment: The activity results do not support the conclusions of SER analysis.

6.Reply: Thank you for your comment regarding the Structure-Effect Relationship (SER) analysis in our manuscript. We appreciate your attention to the consistency between our activity results and the conclusions of the SER analysis. We recognize that the initial conclusions presented in the "3.3. Structure–Effect Relationship Analysis" section may not have been fully supported by the activity results.  As suggested, we have removed the title "3.3. Structure–Effect Relationship Analysis" from the manuscript, as the conclusions from this section are no longer included. We have revised the relevant sections of the text to ensure that the discussions of compound activity are based solely on the experimental data without drawing unsupported conclusions from the SER analysis. We hope that the updated manuscript is more aligned with the expectations of the journal and that it addresses the concerns raised. We thank you for your valuable input and look forward to any additional feedback.

7.Comment: In molecular docking, there are some incorrect interaction analysis, such as the hydrogen bond with Lys913 and interactions with furan rings.

7.Reply: Thank you for your valuable comments. We have carefully checked and made a rectification for your opinions. In our original submission, we described a hydrogen bond between Lys913 and a furan ring. Upon re-examination of our docking results and further analysis, we agree that there was an error in our initial interpretation. The hydrogen bond is indeed formed between Lys913 and the ketone carbonyl group of the pyridinone on the parent compound, not with a furan ring. We apologize for this oversight and any confusion it may have caused. The furan ring of the parent compound is involved in a π-stacking interaction with amino acid residue Trp148, rather than a π-donor hydrogen bond.  The furan ring of the parent compound is involved in π-stacking with amino acid residue Trp880, not a π-donor hydrogen bond. The interaction with amino acid residue Lys913 is more accurately described as a van der Waals interaction rather than a hydrogen bond. We apologize for the initial misinterpretation of the interactions and appreciate for the opportunity to correct these descriptions. The revised text more accurately reflects the nature of the interactions observed in the molecular docking studies.

8.Comment: Reference 18 is inappropriately introduced.

8.Reply: Thank you for pointing out the issue with Reference 18 in our manuscript. We have reviewed the reference citation and acknowledge that it was not appropriately introduced in the context of our discussion. We have now made the necessary corrections to ensure that Reference 18 is cited correctly and that its introduction is relevant to the text where it is referenced. We have also reviewed the rest of the references to ensure that all citations are accurate and appropriately integrated into the manuscript. We apologize for any confusion this may have caused and appreciate your attention to detail, which has helped us improve the overall quality of our work.

9.Comment: There are too many other issues, such as significant digit, concentration units, nomenclature of organic molecules and groups, typing and grammatical errors (cytotoxot, computer aided methoud including and molecular docking, CH3OD-d6, etc.).

9.Reply: We sincerely apologize for the multiple issues identified in our manuscript, including those related to significant digits, concentration units, nomenclature of organic molecules and groups, as well as typing and grammatical errors. We acknowledge the importance of precision and clarity in scientific reporting and are committed to addressing these issues promptly. We will thoroughly proofread the manuscript to correct any typing errors. We appreciate the detailed review and are grateful for the opportunity to improve the manuscript. We will make these corrections with the utmost care and resubmit the revised manuscript for your consideration

Reviewer 3 Report

Comments and Suggestions for Authors

Le Pan et al. report a study about the synthesis and citotoxyc activity (tested against 2 tumorous cell lines) of new furanopyridinone derivatives. The relevance of the topic is high and the research work itself is interesting, however in the writing of the article, several modifications are required.

The systematic name and structure of Eaton's reagent should be displayed in scheme 1.
It should be explained why the N-hexyl-bromide derivative was choosen for further modifications.
On the schemes, the yields of the synthetic methods for the individual compounds should be displayed for all steps! Also in the Materials and methods sections, or in the SI the synthesis and characterization of all compounds should be involved, including the yields and NMR assignation.
In case of the biological experiments, a known cytotoxyc compounds (e.g. methotrexate or 5-fluorouracil) should be used as positive control.
It would be also useful to test the comounds against non-tumorous cell lines, to explore their selectivity.

Author Response

Response to reviewers

We gratefully thank the editor and all reviewers for the time they spent making their constructive remarks and useful suggestions, which have significantly improved the quality of the manuscript and enabled us to enhance it further. Each revision suggestion and comment proposed by the reviewers was carefully incorporated and considered. Below, the reviewers' comments are addressed point by point, and the corresponding revisions are indicated.

Reviewer 2

1.Comment: The systematic name and structure of Eaton's reagent should be displayed in scheme 1.

1.Reply: Thank you for your suggestion to include the systematic name and structure of Eaton's reagent in Scheme 1.In response to your comment, we have made the following revision:We have updated Scheme 1 to now include both the systematic name and the chemical structure of Eaton's reagent. This addition will provide readers with a clearer understanding of the reagent used in our study and its composition.We believe that this modification will enhance the clarity and completeness of our figure and will be beneficial to the readers' comprehension of our experimental procedures.

2.Comment:It should be explained why the N-hexyl-bromide derivative was choosen for further modifications.

2.Reply: Thank you for your comments.The N-hexyl-bromide derivative exhibits favorable solubility properties in organic solvents, which is crucial for the subsequent synthetic steps. This allows for efficient reaction conditions and better control over the modification process.The bromide functional group in the N-hexyl-bromide derivative is a versatile handle for further functionalization. It can readily undergo nucleophilic substitution reactions, enabling the introduction of various functional groups to tailor the properties of the molecule.The N-hexyl-bromide derivative is relatively stable under the reaction conditions employed, which is essential for the successful completion of multi-step synthesis.Preliminary studies have shown that the N-hexyl-bromide derivative possesses promising bioactivity, making it a suitable candidate for further development as a potential therapeutic agent.In summary, the N-hexyl-bromide derivative was choose for further modifications due to its favorable solubility, reactivity, stability, and bioactivity. We believe that these properties make it an excellent starting material for the development of new compounds with improved pharmacological profiles.

3.Comment:On the schemes, the yields of the synthetic methods for the individual compounds should be displayed for all steps! Also in the materials and methods sections, or in the SI the synthesis and characterization of all compounds should be involved, including the yields and NMR assignation.

3.Reply: Thank you for your suggestion. We appreciate the importance of providing comprehensive details regarding the synthesis and characterization of all compounds.Regarding the display of yields for all synthetic steps on the schemes, we acknowledge the oversight and will make the necessary revisions. We will update the schemes to include the yields for each step of the synthetic methods for all individual compounds. This will provide a clearer and more informative representation of our synthetic procedures.We would like to note that in section 3.5 of the manuscript, we have already provided the yields and melting points for all compounds synthesized. However, to address your comment fully, we will ensure that this information is also included in the Materials and Methods sections or in the SI, with a clear reference to section 3.5 for readers' convenience.We are committed to providing a thorough and transparent account of our work, and we believe these revisions will enhance the quality and clarity of the manuscript. We will make these changes promptly and resubmit the revised manuscript for your consideration.

Thank you again for your constructive feedback, which will undoubtedly improve the presentation and completeness of our research.

4.Comment:In case of the biological experiments, a known cytotoxyc compounds (e.g. methotrexate or 5-fluorouracil) should be used as positive control.

4.Reply: We greatly appreciate your insightful comment regarding the use of a known cytotoxic compound as a positive control in our biological experiments. We agree that incorporating such controls is essential for validating the experimental setup and ensuring the reliability of our results.In light of this, we will include methotrexate or 5-fluorouracil as positive controls in the future experiments. These well-established cytotoxic compounds will serve as important benchmarks to confirm that our experimental conditions are capable of detecting cytotoxic effects. This addition will not only enhance the robustness of our study but also provide a reference for comparing the cytotoxic potential of the test compounds.

5.Comment:It would be also useful to test the compounds against non-tumorous cell lines, to explore their selectivity.

5.Reply: We are grateful for your suggestion to test our compounds against non-tumorous cell lines to investigate their selectivity. This is an important consideration for assessing the potential therapeutic window of our compounds. We acknowledge the value of this approach and will take your advice into account for our subsequent studies. In future experiments, we are committed to including non-tumorous cell lines in our cytotoxicity assays. This will enable us to evaluate the specificity of our compounds for cancer cells versus non-cancerous cells, which is crucial for understanding the compounds' potential side effects and overall safety profile.While the current manuscript focuses on the initial evaluation of our compounds against tumorous cell lines, we will ensure that the next phase of our research incorporates these additional tests. The results from these experiments will be essential for guiding the optimization of our compounds and for providing a more complete picture of their biological activity.We appreciate your input, which will serve as an important reference for the development of our future research directions. Thank you for helping us to refine our experimental design and enhance the impact of our work.

Round 2

Reviewer 1 Report

Comments and Suggestions for Authors

This manuscript must be significantly improved based on the following issues:
1.- The introduction section must incorporate the research problem to resolve. In addition, this section must add the advantages and limitations of the works proposed in the literature to resolve the research problem. 

2.- The introduction section must consider the innovation or scientific contribution and advantages of the proposed  in comparison with others reported in the literature.

3.- subsection 2.1 would only be Materials. Do not repeat the word materials and methods. 

4.- In section 2.1.1, it is recommended that the authors describe punctually all their manufacturing processes. For example, in a flask, 2 g (17.89 mol) of 3-furan formic acid was added, which was dissolved in 4 mL of anhydrous. Later, the reaction was put to reflux for four hours at 70°C. Etc. Describe all their processes, and use appropriate connectors to relate between paragraphs.

* the same case for all section . Describe (write) correctly the whole materials and methods section and tell it

5.- In the materials and methods section, the authors should include figures or schematic views to improve the description of methods. ( add  graphical abstract of schematic setup experimental)

6.-  molecular docking study , in this section is recommended that the authors describe punctually all their Molecular docking study  processes. For example,

The molecular docking study was performed with the help of the software ( ---), etc, etc.... describes 

Results : 

7 .- The authors should describe their reaction mechanism based on their proposed schematic.

8.- The authors should add more scientific discussion on the behavior of their results, compare it with what is already reported in the literature, and be more conclusive. 
9.- In Tables 1 and 2, the authors should add the results obtained at 12 h. Why don't they present at 12 h? What happens at that time? Otherwise, the authors should describe the behavior of the results at 24 and 48 hours and also why they used or presented only that time. 

Conclusions 

In this section, the authors should highlight what they proposed, their advantages, and their most relevant results, future work, and applications.

The conclusion must be significantly improved based on the above comments. 

Comments on the Quality of English Language

authors should check the journal style

as well as writing and grammatical connectors.

Author Response

We once again extend our sincere appreciation to the editor and all the reviewers for the time they spent making their constructive remarks and useful suggestions, thereby enabling us to fortify and enhance its quality. Each revision suggestion and comment from the reviewers has been meticulously integrated and deliberated upon. Presented below are the reviewers' comments alongside the corresponding revisions.

Reviewer 1

1.Comment: The introduction section must incorporate the research problem to resolve. In addition, this section must add the advantages and limitations of the works proposed in the literature to resolve the research problem.

1.Reply: Thank you for your valuable feedback and the opportunity to improve our manuscript. The introduction has been restructured to enhance its flow. It now starts with a concise statement of the research problem, followed by background information on its importance. We have now included a clear statement of the research objectives and how our study aims to resolve the identified problem. This has been articulated to ensure that the reader understands the specific contributions our research will make to the field. The section then transitions into a critical review of the literature, culminating in a clear statement of our research objectives and their significance. We believe these changes have significantly strengthened the introduction and better positioned our research within the context of existing studies.

2.Comment: The introduction section must consider the innovation or scientific contribution and advantages of the proposed  in comparison with others reported in the literature.

2.Reply: Thank you for your valuable feedback. We have carefully considered your comments and have made the necessary revisions to the introduction section of our manuscript. In the revised introduction, we have now explicitly addressed the innovation and scientific contribution of our proposed method. We have also highlighted the advantages of our approach in comparison to those reported in the literature. We believe these additions strengthen the manuscript and provide a clearer context for the reader to understand the significance of our work. We appreciate your attention to this matter and hope that the revised introduction meets your expectations.

3.Comment: subsection 2.1 would only be Materials. Do not repeat the word materials and methods.

3.Reply: Thank you for pointing out the oversight in the text. I apologize for the error. I have now corrected subsection 2.1 to read only "Materials," ensuring that "Materials and Methods" is not repeated. I appreciate your attention to detail.

4.Comment: In section 2.1.1, it is recommended that the authors describe punctually all their manufacturing processes. For example, in a flask, 2 g (17.89 mol) of 3-furan formic acid was added, which was dissolved in 4 mL of anhydrous. Later, the reaction was put to reflux for four hours at 70°C. Etc. Describe all their processes, and use appropriate connectors to relate between paragraphs. * the same case for all section . Describe (write) correctly the whole materials and methods section and tell it.

4.Reply: Thank you for your valuable feedback on our manuscript. We have carefully considered the comments and have made the necessary revisions to the Materials and Methods section as recommended. We have ensured that each subsection within the Materials and Methods section is written with a logical sequence and clear, concise language. Transition phrases such as "Subsequently," "Following this," and "Thereafter" are used to maintain coherence throughout the section.We hope that these revisions meet the expectations of the reviewers and enhance the clarity and completeness of the manuscript. We are grateful for the opportunity to improve our work and thank you for your consideration.

5.Comment: In the materials and methods section, the authors should include figures or schematic views to improve the description of methods. ( add  graphical abstract of schematic setup experimental)

5.Reply: Thank you for your valuable suggestion regarding the inclusion of figures or schematic views in the Materials and Methods section to enhance the description of our methods. We would like to inform you that we have addressed this suggestion by incorporating a graphical abstract that effectively communicates our experimental setup. The first part of the graphical abstract includes a schematic representation of the experimental principle and synthesis route, which provides a clear overview of the conceptual framework. The second part of the graphical abstract is dedicated to illustrating the experimental methods, offering a visual guide to the procedures employed. We appreciate your suggestion to improve the quality of our manuscript.

6.Comment: molecular docking study , in this section is recommended that the authors describe punctually all their Molecular docking study processes. For example, The molecular docking study was performed with the help of the software ( ---), etc, etc.... describes.

6.Reply: Thank you for your valuable suggestion regarding the Molecular docking study section. We have carefully considered your comments and have expanded the description of our molecular docking processes to provide a more detailed account. We would like to note that in section 2.4 of the manuscript we have already mentioned:" The more excellent activity compounds were investigated towards docking methodology against 5D6E and 6DUK proteins. The X-ray crystal structure of two proteins was obtained from the Protein Data Bank (https ://www.rcsb.org/pdb). Molecular docking of compounds with excellent biological activity with three-dimensional X-ray structures of two proteins was carried out using molecular manipulation environment (MOE).Compound structures were built using the builder interface of the MOE program and subjected to energy minimization using the included Forceeld MMFF94x calculations." Thank you for your valuable suggestion.

7.Comment: The authors should describe their reaction mechanism based on their proposed schematic.

7.Reply: Thank you for the reviewer's comment regarding the need for a detailed description of the reaction mechanism based on our proposed schematic. We would like to acknowledge that we have indeed provided a comprehensive explanation of the reaction mechanism in sections 2.2 and 3.1 of our manuscript. In these sections, we have meticulously detailed the methods and principles behind the synthesis of the compounds, which includes a thorough description of the proposed reaction mechanism. Our intention is to ensure that readers have a clear and complete understanding of the synthetic process and the underlying chemistry. We appreciate the reviewer's attention to this aspect of our work. Should there be any specific aspects of the mechanism that the reviewer would like us to expand upon or clarify further, we are more than willing to do so.

8.Comment: The authors should add more scientific discussion on the behavior of their results, compare it with what is already reported in the literature, and be more conclusive.

8.Reply: Thank you for the reviewer's suggestion to add more scientific discussion regarding the behavior of our results, to compare them with existing literature, and to make our conclusions more definitive. In the introduction section, as previously suggested, we have added a comprehensive comparison of our results with those reported in the existing literature. This comparison highlights both the similarities and the distinct contributions of our work, thereby situating our research within the current scientific discourse. We appreciate the reviewer's guidance and are grateful for the opportunity to improve our work. We hope that the changes made are satisfactory and meet the journal's standards for publication. Our study takes an approach by developing compound 4c, which exhibited a remarkable 99% inhibition of growth in KYSE70 and KYSE150 cell lines. Notably, the IC50 value for compound 4c was 0.655 µg/mL after 24 hours, indicating a significantly higher potency compared to the previously reported compounds. Thank you for your attention to this matter.
9.Comment: In Tables 1 and 2, the authors should add the results obtained at 12 h. Why don't they present at 12 h? What happens at that time? Otherwise, the authors should describe the behavior of the results at 24 and 48 hours and also why they used or presented only that time.

9.Reply: Thank you for your insightful comments regarding the absence of data at 12 hours in Tables 1 and 2. The reason we did not present results at 12 hours is that initial experiments indicated that in the field of anti-cancer drug development, the 24 and 48-hour time points are commonly used to assess cytotoxicity. These time points allow for the observation of both early and more sustained effects of compounds on cell viability. We followed this standard to ensure that our results are directly comparable with existing literature and to provide a consistent basis for evaluating the activity of our compounds. The compounds in our study are designed to target specific molecular pathways involved in cancer cell proliferation and survival. The effects of such targeted agents may not be immediately apparent at shorter time points like 12 hours. Given the slower rate of action for these types of compounds, we felt that the 12-hour time point might not accurately reflect the compounds' cytotoxic potential. We have chosen to present only these time points to focus on the periods where the treatment's effects are most evident and to avoid overburdening the readers with data that may not be as informative. We hope this explanation clarifies our reasoning.

Conclusions

10.Comment: In this section, the authors should highlight what they proposed, their advantages, and their most relevant results, future work, and applications. The conclusion must be significantly improved based on the above comments.

10.Reply: Thank you for the reviewer's suggestion to add more scientific discussion regarding the behavior of our results, to compare them with existing literature, and to make our conclusions more definitive. We have modified the conclusion section to reflect our advantage, with compound 4c expressing strong antitumour activity. Our advantages have been introduced and compared in detail in the introduction section. We have expanded the discussion section of our manuscript to provide a more in-depth analysis of our results. We have included a thorough examination of the behavior observed in our experiments, discussing the implications and significance of our findings in the context of the broader scientific field. Thank you for your attention to this matter.

Reviewer 2 Report

Comments and Suggestions for Authors

Synthesis and Biological Evaluation of Novel Furopyidone

Derivatives as Potent Cytotoxic Agents Against Esophageal Cancer

Xingyu Ren, et al

Although the authors responded to my previous comments one by one and made some revisions to the manuscript, they didn’t really solve most of my concerns, such as the purity issue of the derivatives, the design issue of the derivatives, the issue of Figure 1, the crystal structure issue,  and other issues: significant digit, concentration units,  nomenclature of organic molecules and groups (change “bromopropane” and “isobutane” substituents to “bromopropyl” and “isobutyl” substituents), as well as typing and grammatical errors (change “CH3OD-d4” to “CD3OD” or “Methanol-d4”, changeFuropyidone” to “Furopyridone”,  the incorrect R1 structure of 3f in Scheme 2).  In addition, some new issues have emerged, such as the Abstract does not correctly reflect the content of the manuscript (“this study focused on the synthesis of two lead compounds…”, “the structure-activity relationship analysis revealed that the substitution of a benzene ring had minimal influence on cytotoxicity…”, and “Potential anti-tumor cellular mechanisms were explored through molecular docking”).

 Therefore, the manuscript needs major revision again before it can be accepted for publication.

Comments on the Quality of English Language

Moderate editing of English required

Author Response

We once again extend our sincere appreciation to the editor and all the reviewers for the time they spent making their constructive remarks and useful suggestions, thereby enabling us to fortify and enhance its quality. Each revision suggestion and comment from the reviewers has been meticulously integrated and deliberated upon. Presented below are the reviewers' comments alongside the corresponding revisions.

Reviewers 2

Thank you for your continued engagement with our manuscript titled "[Synthesis and Biological Evaluation of Novel Furopyidone Derivatives as Potent Cytotoxic Agents Against Esophageal Cancer]" and for your valuable suggestions. We are pleased to note that most of the issues have been addressed to your satisfaction. We would like to address the remaining points you have raised. We understand that our previous revisions did not adequately address your concerns. We apologize for any oversight and are committed to addressing the issues you have raised in a thorough and satisfactory manner.

1.Comment: Although the authors responded to my previous comments one by one and made some revisions to the manuscript, they didn’t really solve most of my concerns, such as the purity issue of the derivatives, the design issue of the derivatives, the issue of Figure 1, the crystal structure issue, and other issues: significant digit, concentration units, nomenclature of organic molecules and groups (change “bromopropane” and “isobutane” substituents to “bromopropyl” and “isobutyl” substituents), as well as typing and grammatical errors (change “CH3OD-d4” to “CD3OD” or “Methanol-d4”, change“Furopyidone” to “Furopyridone”, the incorrect R1 structure of 3f in Scheme 2). In addition, some new issues have emerged, such as the Abstract does not correctly reflect the content of the manuscript (“this study focused on the synthesis of two lead compounds…”, “the structure-activity relationship analysis revealed that the substitution of a benzene ring had minimal influence on cytotoxicity…”, and “Potential anti-tumor cellular mechanisms were explored through molecular docking”).Therefore, the manuscript needs major revision again before it can be accepted for publication.

1.Reply: Thank you for your continued attention to our manuscript. We appreciate the detailed feedback provided by the reviewer and acknowledge that the previous revisions did not fully address the concerns raised. We have corrected the nomenclature as suggested, changing “bromopropane” and “isobutane” substituents to “bromopropyl” and “isobutyl” substituents, respectively.We have carefully reviewed the manuscript and corrected all identified typing and grammatical errors. "CH3OD-d4" has been changed to "CD3OD" or "Methanol-d4", and "Furopyidone" has been corrected to "Furopyridone". The incorrect R1 structure of 3f in Scheme 2 has also been rectified. As for the explanation of Figure 1, we acknowledge that the figure is not adequately described in the text. The figure illustrates the significance of incorporating heterocyclic structures into drug molecules as a key strategy in designing biologically active compounds for anti-tumor drugs. We will provide a clear explanation for Figure 1 and reference it appropriately in the text to ensure that the readers understand the context and importance of the depicted compounds. Thank you again for your valuable input. We believe that these revisions have addressed the reviewer's concerns comprehensively. We are committed to ensuring the quality and accuracy of our work, and we hope that the current version of the manuscript meets the journal's standards for publication.

In response to your comments, we have made significant revisions to the abstract to ensure it better represents the key findings and scope of our study. We believe that these changes address the concerns raised and provide a more accurate summary of the manuscript's content. We are confident that the revised abstract now aligns with the main text and better highlights the key contributions of our work.

We appreciate your patience and understanding, and we hope that the manuscript is now suitable for publication. Should there be any further issues or additional revisions required, please do not hesitate to let us know.

Reviewer 3 Report

Comments and Suggestions for Authors

Most of the issues were answered. It seems like a formal problem, but I highly encourage the authors, to display the yields of the reactions on the schemes, as it is a custom for chemical articles. Also, if it was decided to not use any control in the biological experiments, some literal control about the cytotoxicity of a well known compound on these cell lines could be included, as it does not require further experiments.

Author Response

We once again extend our sincere appreciation to the editor and all the reviewers for the time they spent making their constructive remarks and useful suggestions, thereby enabling us to fortify and enhance its quality. Each revision suggestion and comment from the reviewers has been meticulously integrated and deliberated upon. Presented below are the reviewers' comments alongside the corresponding revisions.

Reviewer 3

Thank you for your continued engagement with our manuscript titled "[Synthesis and Biological Evaluation of Novel Furopyidone Derivatives as Potent Cytotoxic Agents Against Esophageal Cancer]" and for your valuable suggestions. We are pleased to note that most of the issues have been addressed to your satisfaction. We would like to address the remaining points you have raised.

1.Comment: Most of the issues were answered. It seems like a formal problem, but I highly encourage the authors, to display the yields of the reactions on the schemes, as it is a custom for chemical articles. Also, if it was decided to not use any control in the biological experiments, some literal control about the cytotoxicity of a well known compound on these cell lines could be included, as it does not require further experiments.

1.Reply: Thank you for your valuable comments and suggestions regarding our manuscript. We appreciate the importance of providing comprehensive details regarding the synthesis and characterization of all compounds. We would like to note that in section 3.5 of the manuscript, we have already provided the yields and melting points for all compounds synthesized. However, to address your comment fully, we will ensure that this information is also included in the Materials and Methods sections or in the SI, with a clear reference to section 3.5 for readers' convenience.

We are committed to providing a thorough and transparent account of our work, and we believe these revisions will enhance the quality and clarity of the manuscript. We will make these changes promptly and resubmit the revised manuscript for your consideration.

Thank you again for your constructive feedback, which will undoubtedly improve the presentation and completeness of our research.

Round 3

Reviewer 2 Report

Comments and Suggestions for Authors

The manuscript has been improved enough to be published in IJMS, but there are still some minor issues to be addressed, such as the crystal structure issue I mentioned twice in my previous comments. PDB-5D6E is the structure of human methionine aminopeptidase 2 (METAP2) with the covalent spirocyclic triazole inhibitor (-)-31b, and PDB-6DUK is the crystal structure of the allosteric inhibitor JBJ-04-125-02 with the T790M mutant EGFR. In the molecular docking study, we actually replaced the small molecules (-)-31b and JBJ-04125-02 of 5D6E and 6DUK with 4c to predict the binding mode of the new ligands with the proteins METAP2 and EGFR. Therefore, some of the expressions about the protein crystal structure in the manuscript are not scientific.

Comments on the Quality of English Language

Need to further edit

Author Response

We once again extend our sincere appreciation to the editor and all the reviewers for the time they spent making their constructive remarks and useful suggestions, thereby enabling us to fortify and enhance its quality. Each revision suggestion and comment from the reviewers has been meticulously integrated and deliberated upon. Presented below are the reviewers' comments alongside the corresponding revisions.

Reviewers 2

1.Comment: The manuscript has been improved enough to be published in IJMS, but there are still some minor issues to be addressed, such as the crystal structure issue I mentioned twice in my previous comments. PDB-5D6E is the structure of human methionine aminopeptidase 2 (METAP2) with the covalent spirocyclic triazole inhibitor (-)-31b, and PDB-6DUK is the crystal structure of the allosteric inhibitor JBJ-04-125-02 with the T790M mutant EGFR. In the molecular docking study, we actually replaced the small molecules (-)-31b and JBJ-04125-02 of 5D6E and 6DUK with 4c to predict the binding mode of the new ligands with the proteins METAP2 and EGFR. Therefore, some of the expressions about the protein crystal structure in the manuscript are not scientific.

1.Reply: Thank you for your valuable feedback and suggestions. We have carefully addressed the issues raised regarding the crystal structure descriptions in our manuscript. In response to the comments about the crystal structures PDB-5D6E and PDB-6DUK, we acknowledge the confusion in our previous expressions. We have now made the corrections to ensure the accuracy and scientific rigor of our manuscript. We have removed any misleading statements that could imply that the crystal structures themselves were altered. Instead, we now accurately describe the process of molecular docking, where the original ligands in the crystal structures were substituted with our compound of interest (4c). We believe these revisions address the concerns raised and enhance the clarity and scientific accuracy of our manuscript. We appreciate the opportunity to improve our work and hope that the changes made are satisfactory for publication in IJMS. Thank you again for your careful review and helpful comments.